# Locust Bean Gum Nano-Based Hydrogel for Vaginal Delivery of Diphenyl Diselenide in the Treatment of Trichomoniasis: Formulation Characterization and In Vitro Biological Evaluation

**DOI:** 10.3390/pharmaceutics14102112

**Published:** 2022-10-03

**Authors:** Fernanda Padoin dos Reis, Graziela Vargas Rigo, Cristina Wayne Nogueira, Tiana Tasca, Marcel Henrique Marcondes Sari, Letícia Cruz

**Affiliations:** 1Laboratório de Tecnologia Farmacêutica, Programa de Pós-Graduação em Ciências Farmacêuticas, Universidade Federal de Santa Maria, Santa Maria 97105-900, RS, Brazil; 2Grupo de Pesquisa em Tricomonas, GPTrico, Faculdade de Farmácia e Centro de Biotecnologia, Universidade Federal do Rio Grande do Sul, Porto Alegre 90610-000, RS, Brazil; 3Laboratório de Síntese, Reatividade e Avaliação Farmacológica e Toxicológica de Organocalcogênios, Centro de Ciências Naturais e Exatas, Universidade Federal de Santa Maria, Santa Maria 97105-900, RS, Brazil

**Keywords:** nanoparticles, selenium, vaginal route, semisolids, *Trichomonas vaginalis*

## Abstract

Trichomoniasis is the most common nonviral sexually transmitted infection in the world, but its available therapies present low efficacy and high toxicity. Diphenyl diselenide (PhSe_2_) is a pharmacologically active organic selenium compound; however, its clinical use is hindered by its lipophilicity and toxicity. Nanocarriers are an interesting approach to overcome the limitations associated with this compound. This study designed and evaluated a vaginal hydrogel containing PhSe_2_-loaded Eudragit^®^ RS100 and coconut oil nanocapsules for the treatment of trichomoniasis. Nanocapsules presented particle sizes in the nanometric range, positive zeta potential, a compound content close to the theoretical value, and high encapsulation efficiency. The nanoencapsulation maintained the anti-*Trichomonas vaginalis* action of the compound while improving the scavenger action in a DPPH assay. The hydrogels were prepared by thickening nanocapsule suspensions with locust bean gum (3%). The semisolids maintained the nanometric size of the particles and the PhSe_2_ content at around the initial concentration (1.0 mg/g). They also displayed non-Newtonian pseudo-plastic behavior and a highly mucoadhesive property. The chorioallantoic membrane method indicated the absence of hemorrhage, coagulation, or lysis. The compound, from both non-encapsulated and nano-based hydrogel delivery systems, remained on the surface of the bovine vaginal mucosa. Therefore, the formulations displayed the intended properties and could be a promising alternative for the treatment of trichomoniasis.

## 1. Introduction

Trichomoniasis, caused by *Trichomonas vaginalis,* is the most prevalent non-viral sexually transmitted infection (STI) worldwide [1,2]. Infection has been associated with adverse outcomes of pregnancy, increased risk of HIV acquisition and transmission [3], and cervical and prostate cancer [4]. Currently, pharmacological treatment relies on 5-nitroimidazole class drugs, such as metronidazole, tinidazole, and secnidazole. However, these drugs have known safety issues, and drug resistance has already been described [5]. Thus, new active compounds are urgently needed.

Diphenyl diselenide (PhSe_2_) is a synthetic organic selenium compound which is known to have a wide range of biological activities [6]. Studies have confirmed its antioxidant effect [7,8,9], anti-inflammatory action [10], and antimicrobial action against fungi [11,12], viruses [13] and protozoa [14]. However, despite its pharmacological potential, its toxicity, low bioavailability, and poor-water solubility limit its therapeutic use [15,16,17].

In recent decades, nanotechnology has been widely used in the development of improved pharmaceutical devices [18,19]. For example, polymeric nanocapsules are nanocarriers composed of a polymeric wall and an oily core, which allows the incorporation of lipophilic drugs such as PhSe_2_ [20,21,22]. Previous studies demonstrated that the nanoencapsulation of PhSe_2_ enhanced its pharmacological properties [23,24,25] and mitigated its toxic effects [23,26,27]. However, nanocapsules are obtained as a liquid suspension, which is not suitable for topical application. Concerning vaginal administration, independent of the formulation, the major limitation is the low mucosa form retention, reinforcing the importance of mucoadhesive properties for the vaginal delivery of drugs [28]. Therefore, the incorporation of nanocapsule suspensions in mucoadhesive hydrogels has been shown to be a promising approach [29,30].

Hydrogels are obtained using natural or synthetic polymers [31]. Among them, natural ones are preferred because of their low cost, biocompatibility, biodegradability, and wide availability [32]. Locust bean gum is a non-ionic polysaccharide with gelling properties and great stability (pH, salt concentration, and temperature) [33,34]. It has already been demonstrated that nano-based hydrogels increase drug distribution and mucoadhesion in tissue as well as providing controlled release of the active substance [35,36], thereby improving therapeutic efficacy and patient acceptability [37].

Thus, considering (1) the limitations and resistance related to the conventional treatment for trichomoniasis, (2) the pharmacological properties of PhSe_2_ and the promising results previously obtained by its incorporation into nanostructured systems, and (3) the characteristics of hydrogels as a platform for the therapeutic management of trichomoniasis, this study sought to design and evaluate a vaginal hydrogel containing PhSe_2_-loaded nanocapsules for the treatment of trichomoniasis.

## 2. Materials and Methods

### 2.1. Chemicals and Reagents

If not otherwise indicated, all reagents were obtained from Sigma-Aldrich (São Paulo, Brazil). Coconut oil (CO) was bought from Thera Herb (Niterói, Brazil). Eudragit^®^ RS100 (Röhm Pharma, Darmstadt, Germany) was a gift from Almapal (São Paulo, Brazil). HPLC-grade methanol was acquired from Tedia (Rio de Janeiro, Brazil). Locust bean gum was kindly donated by CP Kelco (Limeira, Brazil). Tween 80^®^ (polysorbate 80) was obtained from Delaware (Porto Alegre, Brazil). All other solvents and reagents were of analytical grade. The 30236 *T. vaginalis* isolate (JH 31A #4 from ATCC, Baltimore, MD, USA) was used.

The PhSe_2_ was synthesized based on a previously published method [38] and characterized regarding its chemical purity (99.9% gas chromatography-mass spectrometry/Shimadzu QP2010PLUS GC/MS) and structure (The ^1^H and ^13^C nuclear magnetic resonance/Bruker Avance^TM^ III HD, Billerica, MA, USA).

### 2.2. Preparation of Nanocapsule Suspensions

The nanocapsule suspensions were prepared using the traditional method of interfacial deposition of preformed polymer [39]. To obtain the suspensions, 0.01 g PhSe_2_, 0.077 g Span^®^ 80, 0.1 g Eudragit RS 100^®^, and 330 µL CO were weighed, dissolved in 27 mL acetone (organic phase) and kept under magnetic stirring at 40 °C until total solubilization of constituents had occurred. Then, this phase was injected into a Tween^®^ 80 aqueous dispersion (0.077 g; 53 mL of water) and maintained under moderate magnetic stirring for a further 10 min. Next, the acetone and excessive water were removed by evaporation under low pressure until a final volume 10 mL was obtained, corresponding to a PhSe_2_ concentration of 1 mg/mL (NC PhSe_2_). Placebo formulations (NC B) were also produced for comparative purposes.

### 2.3. Characterization of Nanocapsule Suspensions

The pH of the formulations was assessed by potentiometry using a pH electrode (mPA-210p-MS Tecnopon, Piracicaba, São Paulo, Brazil). The average diameter and polydispersity index (PDI) were determined by photon correlation spectroscopy (1:500 dilution in ultrapure water), while zeta potential was evaluated by the microelectrophoresis technique (1:500 dilution in 10 mM NaCl) (both in Zetasizer Nanoseries^®^, Malvern Instruments, Malvern, Worcestershire, UK).

The total compound content in the formulation was assessed by its extraction from the nanocapsules using a dilution prepared with acetonitrile and acetone (1:1, *v*/*v*) and sonication (10 min). To determine the encapsulation efficiency, an aliquot of the samples was placed in a 10,000 MW centrifugal filter device (Amicon^®^ Ultra, Millipore, Burlington, MA, USA) and the free drug was separated from the formulation using the ultrafiltration/centrifugation technique (2200× *g* for 10 min). The encapsulation efficiency (%) was calculated as the difference between total and free concentrations of PhSe_2_, as determined in the nanostructures and ultrafiltrate, respectively. For compound detection, an analytical method employing a HPLC system was used, following a previously described methodology [22,23]. Quantification was performed on a LC-10A HPLC system (Shimadzu, Kyoto, Japan) equipped with a LC-20AT pump, an UV-VIS SPD-M20A detector, a CBM-20A system controller, and a SIL-20A HT valve sample automatic injector. Separation was achieved using a C_18_ column (Phenomenex Gemini reversed phase, 5 µm, 110 Å, 150 mm × 4.60 mm) at room temperature (25 ± 2 °C). The isocratic mobile phase consisted of methanol and ultrapure water (90:10, *v*/*v*) at 1 mL/min. The injection volume was 20 µL, and the detection wavelength was 243 nm. Before injecting into the HPLC system, all samples were filtered through a 0.45 µm membrane.

### 2.4. In Vitro Evaluation of Biological Properties

In vitro tests to determine the potential cytotoxicity against *T. vaginalis* and antioxidant activity were performed. The concentrations tested were selected based on previous studies [22,23,36] and considering pilot batches to validate the experimental conditions of the assays.

#### 2.4.1. Anti-*T. vaginalis* Action

In this study, *T. vaginalis* 30236, isolated from the American Type Culture Collection (ATCC), was used. The isolate was cultured in vitro in trypticase-yeast extract-maltose (TYM) medium, pH 6.0, supplemented with 10% (*v*/*v*) heat-inactivated serum incubated at 37 °C [40]. Trophozoites with more than 95% viability and normal morphology were used for the assay. Non-encapsulated and nanoencapsulated PhSe_2_ was first diluted with TYM medium or dimethyl sulfoxide (DMSO) to achieve the desired concentrations (8.0 to 32 µg/mL) in a 96-well plate. The same dilutions were prepared to test the blank formulation. Subsequently, the parasite suspension was added to each well (2.0 × 10^5^ trophozoites/mL) and the microplates were incubated for 24 h at 37 °C with 5% CO_2_. After incubation, trophozoite viability was estimated by parasite counts using a hemocytometer with the trypan blue exclusion method (0.2%). The results were expressed as living organisms compared to untreated parasites. The IC_50_ values were calculated by a nonlinear regression model.

#### 2.4.2. Antioxidant Potential

The antioxidant potential was assessed using the DPPH radical scavenging assay [41]. The assay was performed in a 96-well plate using non-encapsulated and nanoencapsulated PhSe_2_ at a concentration range of 1.0 µg/mL to 25.0 µg/mL. The samples were diluted in DMSO or distilled water to achieve the desired concentrations for testing. The DPPH stock solution was prepared by dissolving the DPPH in 50 µM methanol. After 30 min of light protected incubation, the absorbance of each sample was measured using a UV/Vis spectrophotometer (Shimadzu, Kyoto, Japan) at 517 nm. Ascorbic acid was used as positive control in the same concentration range (1.0 to 25 µg/mL). The radical scavenging activity was expressed as the percentage of scavenging capacity based on the negative control (pure DPPH reagent), as previous described [22].

### 2.5. Hydrogels Preparation

The hydrogels were prepared using a glass mortar and pestle in which the placebo or compound-loaded nanocapsule suspension (10 mL) was gradually added to 0.3 g of the Locust bean gum (3%), i.e., the gel-forming agent (HG NC PhSe_2_ or HG NC B). For the hydrogel containing non-encapsulated PhSe_2_ (HG PhSe_2_), the compound was first solubilized in a mixture of 330 µL CO and 670 µL DMSO and added to 9 mL of previously prepared aqueous gum dispersion. Both hydrogels containing the compound presented a final PhSe_2_ theoretical concentration of 1 mg/g. No further components were added to the formulations (pH corrective, antioxidants, and antimicrobial preservatives). The hydrogels were stored in plastic containers at room temperature (25 °C).

### 2.6. Hydrogel Characterization

After preparing the hydrogels, the general characteristics of the formulations were analyzed, i.e., macroscopic aspect, pH, particle size, compound content, spreadability, and rheological behavior.

The pH values were determined in a hydrogel dispersion prepared with distilled water (10%, *w*/*v*) using a calibrated potentiometer (mPA-210p-MS Tecnopon, Piracicaba, São Paulo, Brazil). To determine the average diameter, the hydrogels were dispersed in ultrapure water 1:500 (*w*/*v*), filtered through a quantitative filter (45 μm), and analyzed by photon correlation spectroscopy (Zetasizer, Malvern Instruments, Malvern, Worcestershire, UK). Lastly, the total quantity of PhSe_2_ in the hydrogels was determined employing the above-mentioned analytical procedure (see Section 2.2). Then, 0.01 g of hydrogels was mixed with acetonitrile:acetone (1:1, *v*/*v*) and magnetically stirred for 30 min before undergoing 30 min of sonication in a ultrasonic bath. Then, the samples were filtered through a membrane (0.45 µm) and injected into the equipment.

The spreadability of the hydrogels was assessed using the parallel plate method [42,43]. An amount of hydrogel samples was placed in the central hole (1.2 cm in diameter) of a mold glass plate, which was positioned on the scanner surface (HP Officejet, model 4500 Desktop). The mold was carefully removed, and the sample was submitted to a cumulative pressure provided by a conjunct of glass plates. The plates had a known weight and were cumulatively added each 1 min. To determine the spreadability profile, an image of each spread area was captured for quantification, using the software ImageJ (Version 1.49q, National Institutes of Health, Bethesda, MD, USA), and for spreadability factor (Sf) estimation (Equation (1)). The following equation was used to determine the spreading capacity on a smooth horizontal surface when one gram of weight was added:Sf = A/W(1)

The rheological behavior of hydrogels was determined employing a Brookfield viscometer (RVDV-I-PRIME model, Brookfield, Middleboro, MA, USA) with a RV06 spindle. An aliquot of HGs was submitted to a speed range from 2 to 100 rpm at room temperature (25 ± 1 °C). To better elucidate the rheological behavior of the hydrogels, the experimental data were fitted to the mathematical models of Bingham (Equation (2)), Casson (Equation (3)), and Ostwald (Equation (4)).
τ = τ0 + ηγ(2)
τ^0.5^ = τ0^0.5^ + η^0.5^ γ^0.5^(3)
τ = κγ^n^(4)
where τ0 is the yield stress, η is the viscosity, n is the index of flow, κ is the index of consistency, τ is the shear stress, and γ is the shear rate.

### 2.7. Mucoadhesion and Mucosa Permeation Evaluations

#### 2.7.1. Nanocapsule Formulations

The mucoadhesive potential was evaluated based on a mucin-particle method [44]. Mucin porcine Type II was suspended in 0.1% ultrapure water (*w*/*v*). The liquid suspensions were diluted in the mucin solution (1:500, *v*/*v*), and the mean particle size and zeta potential were determined in Zetasizer.

#### 2.7.2. Hydrogel Formulations

To evaluate the mucoadhesive properties and permeation/penetration, cow vaginal mucosa was employed. Tissue samples were obtained from a local slaughterhouse (Frigorífico Silva, Santa Maria, RS, Brazil). The mucosa was carefully separated from adjacent tissues and cut to appropriate sizes for both experiments. The fragments were kept in a freezer at −18 °C.

The mucoadhesive properties of the hydrogels were evaluated on a modified two-arm physical balance, based on a previous study [45]. For comparative purposes, carbopol hydrogel was prepared and evaluated regarding its mucoadhesive strength. At one end of the physical balance, a falcon tube was used to receive water volumes. At the other end, the mucosa circle cut section was attached to a rubber stopper, in contact with the hydrogel sample. An aliquot of the hydrogels (0.8 g) was placed under the mucosa and contact was ensured for 60 s under 1 N force. Subsequently, the volume of water required to detach the film from the skin surface was used to calculate the mucoadhesive strength, as shown in the equation below:MS (dyne/cm²) = (Weight of detachment × Gravity acceleration [980 cm/s^2^])/Area of tissue exposed (4.5 cm^2^)(5)

The permeation/penetration of PhSe_2_ in the hydrogels was measured using vertical Franz diffusion cells at 37 ± 0.5 °C (*n* = 6) and cow vaginal mucosa as a biological barrier. The receptor medium consisted of phosphate buffer with a pH of 5.5, which was kept under magnetic stirring. An amount of non-encapsulated or nanoencapsulated PhSe_2_ hydrogels was applied to the mucosa surface and incubated for 8 h. Then, a medium aliquot was collected from the receptor compartment to determine if the drug had permeated the membrane. The mucosa was removed and chopped into smaller fragments. The drug extraction was performed with 5 mL methanol, 3 min of vortexing and 15 min in an ultrasonic bath. Finally, 50 µL of sample was filtered through a 0.45 µm cellulose membrane and injected into the HPLC system, using the method described in Section 2.2. The results were expressed as µg of PhSe_2_/cm^2^ of mucosa.

### 2.8. Evaluation of Irritant Potential of the Formulations

The hen’s egg test chorioallantoic membrane (HET-CAM) was used to estimate the irritant potential of the formulations. This evaluation was approved by the Ethics Committee on the Use of Animals of the Federal University of Santa Maria (CEUA) (Process No. 5428271020/2021, 01/05/2021). Fertilized hen’s eggs with 10 days of incubation (36 ± 0.5 °C and 88% relative humidity), donated by Languiru (Teutonia, Brazil), were used. During the test, the most external shell and the white membrane were removed without damaging the inner membrane, and an amount of sample (300 µL for nanocapsules and 0.3 g for hydrogels) was placed onto distinct chorioallantoic membranes (CAM) (*n* = 3/formulation). After 20 s, the samples were removed with saline solution and the CAM was monitored for 300 s. During this period, the onset of the vasoconstriction phenomena, hemorrhage, or coagulation was recorded. The formulations were compared with the bulk compound (aqueous solution with 10% of Tween^®^ 80 and 10% of DMSO). Positive (0.1 M NaOH) and negative (0.9% NaCl) controls were also measured. Each test was performed in triplicate and the mean score of three eggs was used. The irritation scores (IS) were calculated using Equation (6):(6)$$\mathrm{IS}=\frac{301-h}{300}\times 5+\frac{301-v}{300}\times 7+\frac{301-c}{300}\times 9$$
where h = hemorrhage time; v = vasoconstriction time, and c = coagulation time. From the IS values obtained, the lesions were classified as non-irritant (0–0.9), slight (1–4.9), moderate (5–8.9), or severe irritant (9–21).

### 2.9. Data Presentation and Statistical Evaluation

The results are expressed as mean ± standard deviation (S.D.). Unless otherwise stated, all experiments were conducted in triplicate, with at least two independent experimental batches. Data normality was estimated using the D’Agostino and Pearson omnibus normality tests. Statistical comparisons among experimental groups were performed using unpaired and paired Student’s t test or ordinary one-way analysis of variance (ANOVA). The Tukey’s test for post hoc comparison was applied when appropriate. The GraphPad Prism^®^ statistical software version 7 (San Diego, CA, USA) was used for the statistical analysis of experimental data and to create the figures. Values of *p* < 0.05 were accepted as statistically significant for all comparisons.

## 3. Results

### 3.1. General Characterization of Nanocapsule Suspensions

After preparation, all formulations macroscopically showed a homogeneous “milk-like” appearance without any visible precipitation. The NC PhSe_2_ suspension had a slightly yellowish color due to the compound loading in the nanostructures. A cloudy appearance and opalescent reflection characteristic of colloidal particles were observed as well.

Both placebo and PhSe_2_-loaded nanocapsule suspensions had an average particle diameter in the nanometric range (206 ± 20 nm and 193 ± 14 nm, respectively), PDI values of around 0.2 (0.22 ± 0.05 nm and 0.19 ± 0.07, respectively), positive zeta potential values (+8.50 ± 2.22 mV and +6.86 ± 0.66 mV, respectively), and pH in the acidic range (4.97 ± 0.18 and 5.04 ± 0.02, respectively). The total PhSe_2_ content in the nanocapsule formulations was close to the theoretical value (1.04 ± 0.03 mg/mL), and a high encapsulation efficiency was achieved (99.92%) (see Table 1). Statistical comparisons revealed no significant differences between any of the evaluated parameters (*p* > 0.05).

### 3.2. Anti-T. vaginalis Action and Antioxidant Properties of Nanocapsules

The results of the bioassay demonstrated that both unloaded and loaded PhSe_2_ induced a similar reduction in trophozoite viability at the lowest concentration tested (8 µg/mL) (*p* > 0.05) (Figure 1A). At the intermediate concentration (16 µg/mL), nanoencapsulated PhSe_2_ showed superior anti-*T. vaginalis* action in comparison to non-encapsulated compound (*p* < 0.05). For the highest concentration tested (31 µg/mL), the same degree of trophozoite viability reduction was observed as that of NC (PhSe)_2_, while non-encapsulated PhSe_2_ showed increased cytotoxic action against the parasite (*p* < 0.05). Overall, placebo nanocarriers did not modify trophozoite viability (*p* > 0.05). Based on these results, the obtained IC_50_ values for PhSe_2_ and NC PhSe_2_ were 27 µg/mL and 34 µg/mL, respectively.

Regarding the DPPH assay, all tested concentrations demonstrated that the nanoencapsulation improved the scavenger action in comparison to non-encapsulated PhSe_2_ (*p* < 0.05) (see Figure 1B).

### 3.3. General Characterization of Hydrogels

All of the prepared hydrogels presented homogeneous macroscopic appearance, without lumps or precipitates. Both HGs containing PhSe_2_ exhibited a yellowish color; the non-encapsulated PhSe_2_ formulation was bright yellow, while HG NC PhSe_2_ showed an opaquer appearance due to compound loading. The hydrogel prepared with the placebo formulation had a milky white appearance.

The average diameter of the particles in the hydrogel presented a slight increase in comparison to the suspension, but was still within the nanometric size scale. The PDI values were compatible with the homogeneous distribution of particles size, and the pH values of all prepared hydrogel formulations were within the acidic and neutral range. The total content in the hydrogel was 1.08 ± 0.09 mg/g for the non-encapsulated formulation and 0.96 ± 0.01 mg/g for the nano-based hydrogel. Concerning spreadability, all hydrogels showed a proportional increase in the spread area as a consequence of the application of pressure (Figure 2A) and similar spreadability factor values. Statistical comparisons revealed no significant differences among any of the evaluated parameters (*p* > 0.05) (Table 2).

The rheological behavior of the hydrogels was investigated by obtaining viscograms and rheograms (see Figure 2B) through the graphic representation of viscosity versus shear rate, and shear stress versus shear rate, respectively. The data evidenced that the proportion of the shear rate versus shear stress was not linear. The viscosity values varied as a consequence of the applied shear rate variation, suggesting that all prepared hydrogels presented non-Newtonian flow. Furthermore, the mathematical modeling of these data demonstrated that the Ostwald model was the best one to describe the hydrogel, indicating a pseudoplastic behavior (Table 2). The flow (ɳ) and consistency (Ƙ) indexes revealed that both nano-based HG had superior Ƙ index values in comparison to the formulation prepared with non-encapsulated compound (*p* < 0.05) (Table 2).

### 3.4. Mucoadhesion Assessment

After the interaction with mucin, both placebo and NC PhSe_2_ suspensions showed an increase in the mean particle diameter and an inversion in the polymer electric charge for all samples tested when compared to the original values (see Figure 3A). A statistical evaluation performed by paired the Student’s t test revealed a significant difference among the groups (*p* < 0.05). Concerning the mucoadhesive strength of the hydrogels, the results revealed that the nano-based formulations were more mucoadhesive than the hydrogel PhSe_2_ or the carbopol hydrogels (*p* < 0.05) (see Figure 3B).

### 3.5. (PhSe)_2_ Mucosa Permeation

After 8 h of incubation, similar quantities of PhSe_2_ were detected in the cow vaginal mucosa treated with the hydrogels containing non-encapsulated or loaded compound (*p* > 0.05) (Figure 4). No PhSe_2_ was detected in the receptor compartment medium.

### 3.6. HET-CAM Assay

Table 3 shows the IS of all tested samples. No alterations, such as lysis, coagulation, or hemorrhage, were observed in the CAM after exposure to the formulations or the negative control (Figure 5A–O). The positive control was classified as a severe irritant (see Figure 5P).

## 4. Discussion

Despite the worldwide incidence and serious health consequences of trichomoniasis infection, its clinical management is still limited to a few systemic or local antimicrobial therapies, such as metronidazole and tinidazole [46]. Additionally, the increasing prevalence of *T. vaginalis* strains that are resistant to these drugs and infection recurrences indicate that treatment is becoming a challenge, which reinforces the importance of developing new therapeutic alternatives [5]. In this sense, many efforts have been devoted to the development of vaginal drug delivery systems, including the use of nanocarriers [47]. The current study designed a nano-based vaginal hydrogel for PhSe_2_ delivery as a possible treatment for trichomoniasis. The formulation presented anti-*T. vaginalis* and antioxidant in vitro actions, suggesting potential for mitigating the trophozoite replication and oxidative damages caused by the infection. Furthermore, the hydrogels containing nanocapsules showed superior bioadhesion potential, which is a desirable property for formulations intended for vaginal administration.

The nanoencapsulation of PhSe_2_ has already been undertaken in previous studies. In this regard, different polymers (poly-epsilon-caprolactone and Eudragit^®^ RS100), oil cores (canola oil, medium chain triglycerides, and coconut oil) and compound concentrations (1, 1.56, and 5 mg/mL) were applied [22,23,48]. Based on previous studies, we designed a formulation to better fit the requirements for PhSe_2_ vaginal delivery, using a polymeric wall that could provide higher mucoadhesion and a compatible oil core. The data obtained in our research corroborate the findings of later reports, i.e., the nanocapsules had particle sizes within a nanometric range and homogeneous size, as well as compatible compound content values. Moreover, a high encapsulation efficiency was achieved, which could be attributed to the solubility of PhSe_2_ in the oil core (24.08 mg/mL) [22]; this further contributed to system stability. These characteristics were similar to those described with polymeric nanocapsules prepared with poly-epsilon-caprolactone and medium chain triglycerides [23,48].

To developing novel vaginal drug delivery systems, the physiological characteristics of the vaginal tissue must be considered [28]. The formulated nanocapsules presented suitable pH values for their intended route for formulation application (acid range) and great mucoadhesion, as demonstrated using mucin, the main component of mucus. This method is based on the electrostatic interaction between negatively charged mucin (sialic groups) and suspensions with cationic characteristics [49]. Thus, the mucoadhesive property of the formulation may have been due to the cationic polymeric shell of Eudragit^®^ RS100, which creates attraction forces at the interface, due to the charges difference, increasing the mean size and inverting the zeta potential [30]. These data suggest that the polymer plays an important role in increasing interaction with the mucosa. Based on our data regarding the mucin particle method, a putative mechanism for the higher mucoadhesion strength is electrostatic interaction. In the mucin method, the zeta potential inversion results from the contact between sialic groups in mucin, which are negatively charged, with positively charged particles, which is a characteristic of cationic polymers, such as Eudragit^®^ RS 100. In this sense, ionic interaction occurs between the anionic and cationic surface of the particles when they come into contact with each other. Pharmaceutical dosage forms composed of a polymeric bioadhesive material may ensure a prolonged residence time in the vaginal cavity, increasing drug interaction with mucosa, and thereby, improving efficacy [46,50].

In recent decades, several studies on the antimicrobial activity against pathogenic fungi of organoselenium compounds, bacteria, and viruses have been published [51], suggesting the potential of using these molecules to treat infectious diseases. In support of this, preclinical data showed that PhSe_2_ modulates inflammation and oxidative events (see [6]), both of which are pivotal processes in infections. Importantly, to the best of our knowledge, to date, no report has investigated the anti-*T. vaginalis* action of PhSe_2_. In our study, this was demonstrated via non-encapsulated and nanoencapsulated compounds using an in vitro assay. Additionally, our data demonstrated that PhSe_2_-loading into nanocapsules increased its scavenger action in comparison to bulk compound, highlighting the improvements achieved by nanoencapsulation. Notably, the obtained IC_50_ values for both compound forms were within a concentration range in which no considerable cytotoxic effects were detected in different mammalian cell lines [23,52].

Regarding the pathophysiology of *T. vaginalis* infection, the production of reactive oxygen species is considered a critical event for trophozoite signaling pathways. In this sense, reducing the extent of their synthesis could mitigate pathogen survival [53]. Remarkably, Malli and co-workers (2018) showed that mucoadhesive nanocarriers had superior uptake by *T. vaginalis* and increased cytotoxicity [54]. However, even considering that NC PhSe_2_ had excellent mucoadhesive properties, non-encapsulated compound was more efficient at reducing the viability of trophozoites, which may be a consequence of the controlled rate of drug release from the nanocarriers. The partition barrier between the oil core and polymeric wall must be overcome by PhSe_2_ for its release into the external aqueous medium which, on its own, does not provide adequate conditions for its release. While the non-encapsulated compound was immediately available for exposure to *T. vaginalis*, the nanocapsules provided a slow compound partition profile. However, our test confirmed that the drug was still active after encapsulation, and that the formulation still had acceptable activity levels.

Considering our promising results, in order to enable intravaginal administration of the nanostructured suspension, hydrogels were produced, i.e., aqueous semisolid formulations. Hydrogels are sensorially better in terms of their skin perception and are easy to administer, enhancing patient compliance. They can be high mucoadhesive, a desired characteristic for vaginal application [19]. To produce the hydrogels, locust bean gum was used as a gelling agent; this compound is a natural polymer that has already demonstrated effectiveness in food, cosmetic, and pharmaceutical applications, including in gel formulations [33]. Locust bean gum has suitable mechanical and adhesiveness properties, biodegradability, biocompatibility, and low toxicity [33,55]. The nanocapsules thickened with locust bean gum (3%) maintained similar characteristics in comparison to the initial suspension (average diameter, PDI values, and PhSe_2_ content), suggesting that the thickening process did not have a negative impact on the physicochemical properties. These data are in accordance with previous studies by our research group, in which a slight modification in the average diameter or/and PDI values did not impair the stability or pharmacological properties of the hydrogels [24,56].

The vaginal route has pH values within the acid range (3.8–4.5), which is considered to be effective against pathogen proliferation and for mucosa homeostasis [57]. The designed hydrogels presented slightly acid to neutral pH values, and may cause a temporary modification in mucosal pH, similarly to many physiological conditions, such as menstruation, vaginal transudate, and contact with semen [58]. Of particular importance, trichomoniasis also causes an increase in the vaginal pH [59,60]. Notably, vaginal formulations presenting pH values around the neutrality range have been described in the literature for the treatment of bacterial vaginosis and HSV infection [61,62,63]. Therefore, despite the non-optimal pH characteristic of the formulation, it is unlikely to cause significant harm to the mucosa and may contribute to modulating the altered pH while mitigating the infection.

When designing a semisolid formulation, spreadability and the rheological properties are of great importance [42,64]. All of the hydrogels developed in our study presented similar spreadability profiles and Sf, corroborating previous scientific studies that developed semisolid formulations intended for vaginal administration [30,36]. Furthermore, both nano-based and non-nano-based hydrogels exhibited pseudoplastic flow, i.e., the viscosity decreased as the applied shear stress increased, and were mathematically described by the Ostwald model, indicating that they spread under low shear stress. Notably, our nano-based formulation presented superior viscosity because of their higher Ƙ index values in comparison to the hydrogels prepared with non-encapsulated PhSe_2_, which is a desirable characteristic, as it increases both formulation residence time and contact with the tissue. These data indicate that nano-based formulations can strike a proper balance between spreadability and the retention of hydrogels at the application site.

To date, several topical dosage forms have been proposed for vaginal drug delivery, but their effectiveness is limited by rapid removal from the vaginal cavity [32]. Hence, multiple and periodic local administrations are required, increasing side effects and reducing patient compliance [58]. Thus, a desired property of vaginal pharmaceutical dosage forms is mucoadhesion, which ensures a prolonged residence time in the tissue [28,37]. Our nano-based hydrogels presented superior bioadhesive strength than non-encapsulated compound hydrogels and a Carbopol formulation, which is a reference gel-forming polymer for providing mucoadhesion potential to pharmaceutical dosage forms [65]. Additionally, the use of Eudragit^®^ RS100 to compose the polymeric wall of the nanocapsules may increase the mucoadhesion, as demonstrated by the mucin method. Accordingly, previous studies reported enhancemed mucoadhesion for formulations containing nanostructured systems [30,35,36,66]. These data underline the importance of the selection of the formulation components for vaginal hydrogels (cationic polymer and locust bean gum), as longer retention at the application site can be achieved by using mucoadhesive materials [19].

To study the permeation profile of the PhSe_2_ hydrogels, Franz-type diffusion cells and cow vaginal mucosa were used. The choice of this tissue was based on physiological similarities with human vaginal mucosa, such as the impact of hormone regulation on the vaginal epithelium thickness and mucus secretion [65]. After 8 h of exposure, the hydrogels delivered similar quantities of PhSe_2_ to the vaginal mucosa, while none was detected in the receptor medium. The obtained permeation profile was promising because in trichomoniasis, the parasite adheres to the epithelial cells of the vaginal mucosa [67]. In this context, we suggest that the smaller the quantity of PhSe_2_ that permeates the vaginal mucosa, the more effective the treatment would be. Notably, safety is improved by minimizing systemic absorption [15]. Collectively, the permeation profile and mucoadhesive property of HG NC PhSe_2_ provide a desirable combination in order to prolong the local effect of the active compound, revealing the formulation as a promising candidate for trichomoniasis management.

Despite the bright outlook of nano-based formulations, there is increasing concern about the methods to assess their safety [68]. In this sense, we applied the HET-CAM method, a substitutive, semiquantitative methodology, as a preliminary tool for toxicological screening. This approach is widely used to assesses the irritant potential of substances intended for ophthalmic and cutaneous administration, that, according to ISO 10993-10 [69], can be directly correlated to vaginal tissue without animal testing, suggesting a similar epithelia response in terms of irritancy. In our findings, after exposing the membrane to the samples, no signs of irritation were detected, indicating a low toxic potential and suggesting that all tested formulations were safe for vaginal administration.

Lastly, for proper data interpretation, it is important to recognize some limitations in our study. Notably, the highly lipophilic characteristic of PhSe_2_ hindered its investigation at higher concentrations in an in vitro anti-*Trichomonas vaginalis* test. The same physicochemical property prevented us from determining the release profile of the compound from the nanocarriers in vitro. However, even considering these limitations, previous studies and the present data reinforce the potential merits of PhSe_2_ and formulations for multiple pharmacological purposes.

## 5. Conclusions

In this study, we prepared a locust bean gum nano-based hydrogel for vaginal delivery of PhSe_2_. Remarkably, the anti-*T. vaginalis* action of PhSe_2_ was demonstrated for the first time, and was maintained after nanoencapsulating the compound into mucoadhesive Eudragit^®^ RS100 nanocapsules. The PhSe_2_ NC suspension presented the characteristics of a nanostructured system and improved the scavenging potential of the compound. The hydrogels were obtained by an easy and low-cost process, yielding formulations with suitable spreadability and rheological profiles, high mucoadhesive properties, PhSe_2_ mucosal retention, and non-irritating potential. Therefore, HG PhSe_2_ NC can be considered a promising new approach to treat trichomoniasis.

## Figures and Tables

**Figure 1 pharmaceutics-14-02112-f001:**
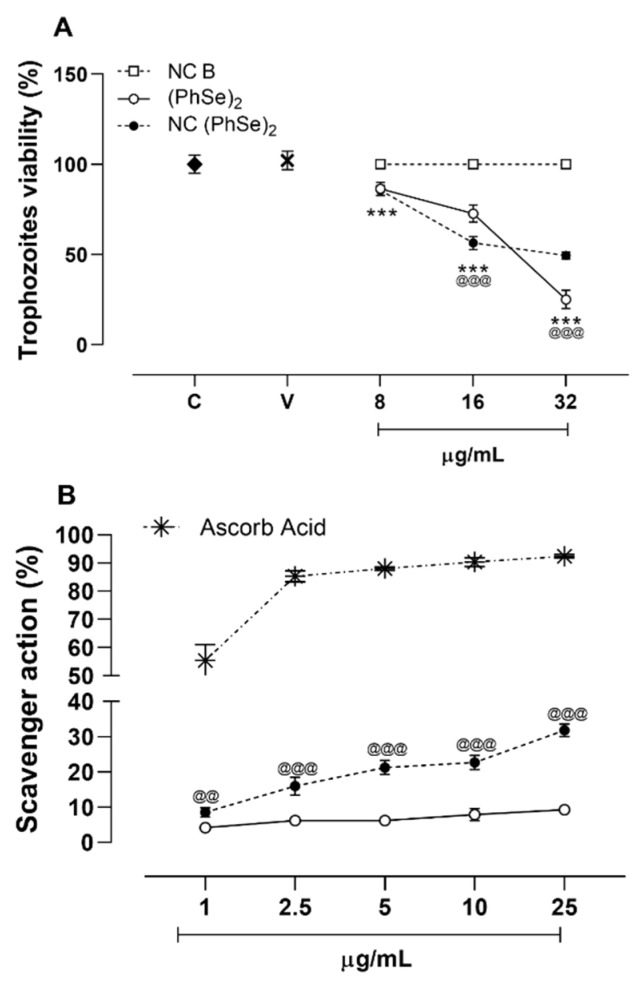
In vitro evaluation of anti-*T. vaginalis* activity against ATCC 30236 isolate (**A**) and DPPH scavenger assay of the formulations (**B**). Data are presented as mean and standard deviation. Statistical evaluations were performed by one-way ANOVA followed by Tukey. (***) *p* < 0.001 denotes statistical difference between control and treatment; @ denotes statistical difference between (PhSe)_2_ and NC (PhSe)_2_; (@@) *p* < 0.01 and (@@@) *p* < 0.001.

**Figure 2 pharmaceutics-14-02112-f002:**
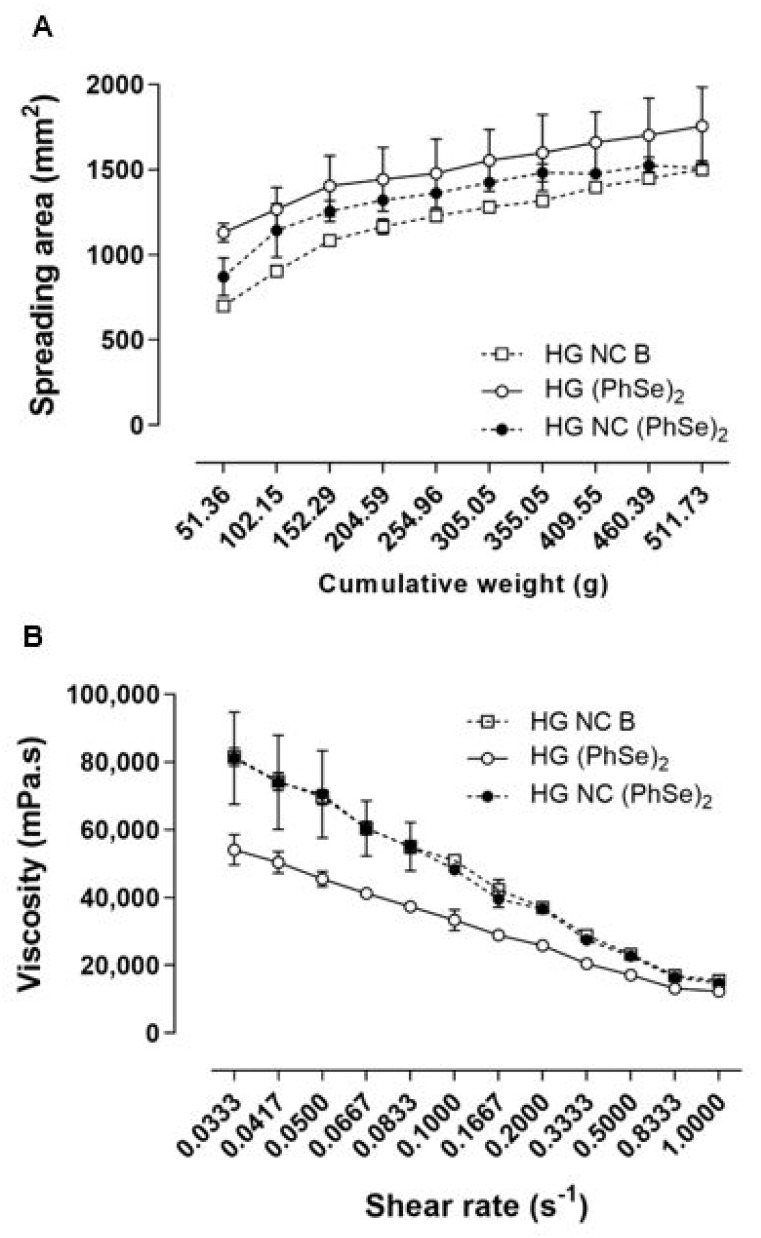
Graphical representation of the spreadability profile (**A**) and viscosity (**B**) of the hydrogels. Data are presented as mean and standard deviation. Statistical evaluations were performed by one-way ANOVA followed by Tukey (*p* > 0.05).

**Figure 3 pharmaceutics-14-02112-f003:**
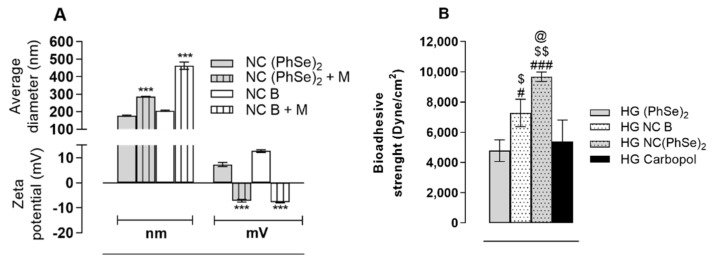
Effect of mucin (+M) treatment on the average diameter and zeta potential of nanocapsules (**A**) and the mucoadhesion strength of hydrogels (**B**). The statistical evaluation was performed by paired Student’s *t* (**A**) or one-way ANOVA followed by Tukey. (***) *p* < 0.001 denotes statistical difference between initial and after mucin exposure values of average diameter and zeta potential; # denotes statistical difference in comparison to HG (PhSe)_2_; (#) *p* < 0.05 and (###) *p* < 0.001; $ denotes statistical difference in comparison to HG Carbopol; ($) *p* < 0.05 and ($$) *p* < 0.001; @ denotes statistical difference between NC (PhSe)_2_ and NC B; (@) *p* < 0.05.

**Figure 4 pharmaceutics-14-02112-f004:**
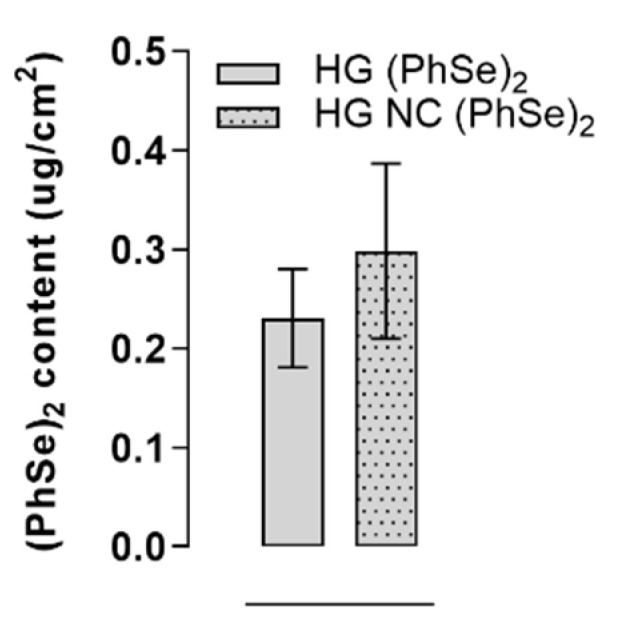
In vitro permeation study in vaginal mucosa after the application of hydrogels. The statistical evaluation was performed by unpaired Student’s *t* (*p* > 0.05).

**Figure 5 pharmaceutics-14-02112-f005:**
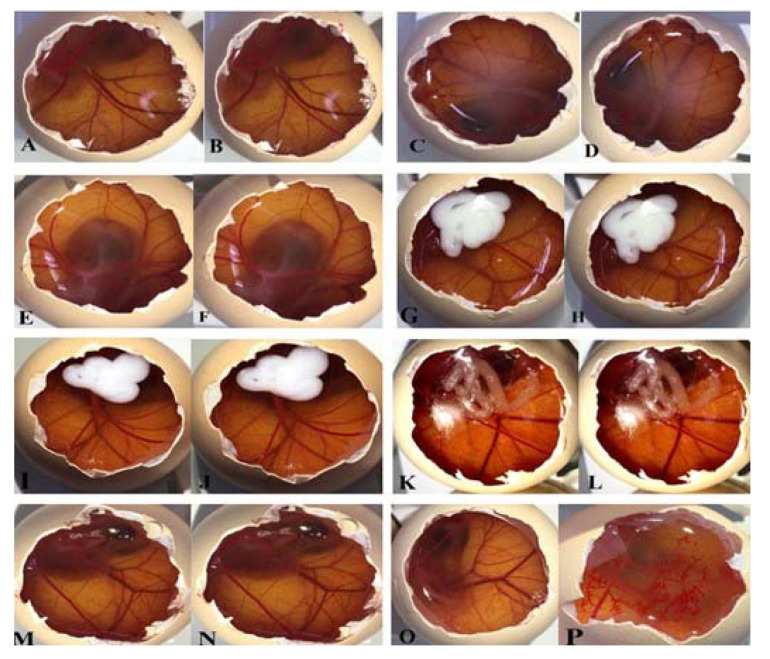
Images of the chorioallantoic membrane test demonstrating CAM exposed when the formulations were applied (time zero) and 5 min v application: (**A**,**B**) PhSe_2_; (**C**,**D**) NC PhSe_2_; (**E**,**F**) NC B; (**G**,**H**) HG NC PhSe_2_; (**I**,**J**) HG NC B; (**K**,**L**) HG PhSe_2_; (**M**,**N**) Locust bean gum HG (vehicle); (**O**,**P**) negative (0.9% NaCl) and positive control (0.1 M NaOH), respectively.

**Table 1 pharmaceutics-14-02112-t001:** Physicochemical characteristics of the developed formulations.

Formulation	AS (nm)	PDI	pH	Drug Content (mg/g)
HG (PhSe)_2_	-	-	7.52 ± 0.10	1.08 ± 0.09
HG NC B	295 ± 24	0.29 ± 0.09	6.16 ± 0.14	-
HG NC (PhSe)_2_	237 ± 4	0.25 ± 0.01	6.20 ± 0.14	0.96 ± 0.01

The results are expressed as S.E.M. of three independent experiments performed in triplicate. All data were analyzed by ordinary one-way ANOVA followed by Tukey’s tests, *p* > 0.05. Abbreviations: AS = Average size; PDI = Polydispersity index.

**Table 2 pharmaceutics-14-02112-t002:** Spreadability profile and rheological behavior concerning the regression coefficients (r^2^) for mathematical modeling in shear rate-shear stress curves to models of Ostwald, Casson, and Bingham.

**Formulation**	**Index**		**Sf (mm^2^/g)**
	**ɳ**	**Ƙ**	
HG (PhSe)_2_	0.565 ± 0.0209	8.08 × 10^4^ ± 7565	3.43 ± 0.44
HG NC B	0.513 ± 0.0114	1.53 × 10^5^ ± 9511 *	2.94 ± 0.02
HG NC (PhSe)_2_	0.503 ± 0.0580	1.55 × 10^5^ ± 4410 *	2.95 ± 0.08
**Formulation**	**Equations**		
	**Bingham**	**Casson**	**Ostwald**
HG (PhSe)_2_	0.982 ± 0.006	0.992 ± 0.004	0.998 ± 0.001
HG NC B	0.968 ± 0.006	0.984 ± 0.004	0.997 ± 0.002
HG NC (PhSe)_2_	0.968 ± 0.006	0.985 ± 0.004	0.997 ± 0.001

Results are expressed as S.E.M. of three independent experiments performed in triplicate. All data were analyzed by ordinary one-way ANOVA. The asterisk denotes a significant difference in comparison to the to the HG PhSe_2_ (*) *p* < 0.05. Abbreviations: Sf = spreadability factor; ɳ = flow index; Ƙ = consistency index.

**Table 3 pharmaceutics-14-02112-t003:** Hen’s Egg Test irritation scores.

Sample	Irritation Score (IS)
(PhSe)_2_	0.00 ± 0.0
NC (PhSe)_2_	0.00 ± 0.0
NC B	0.00 ± 0.0
HG NC (PhSe)_2_	0.00 ± 0.0
HG (PhSe)_2_	0.00 ± 0.0
HG NC B	0.00 ± 0.0
NaCl 0.9%	0.00 ± 0.0
NaOH 0.1 M	13.6 ± 0.1

NaCl (negative control); NaOH (coagulation control).

## Data Availability

Not applicable.

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
