# Peer review of "Locust Bean Gum Nano-Based Hydrogel for Vaginal Delivery of Diphenyl Diselenide in the Treatment of Trichomoniasis: Formulation Characterization and In Vitro Biological Evaluation"

_pharmaceutics, 2022, doi:10.3390/pharmaceutics14102112_

Round 1

Reviewer 1 Report

See the comments attached 

Author Response

Dear reviewer,

We would like to express our thanks for your helpful and constructive comments on the manuscript of “Locust bean gum nano-based hydrogel for diphenyl diselenide vaginal delivery intending trichomoniasis treatment: Formulation characterization and in vitro biological evaluations” (Manuscript ID: pharmaceutics-1914871). We have carefully considered the revision and tried our best to address the suggestions. In the current version of the document, all the changes are highlighted in green throughout the text. Below we provide the point-by-point responses to each of your queries.

Best regards

Letícia Cruz

Reviewer 2 Report

Authors deal with the treatment of Trichomoniasis which, despite it is one of the most prevalent non-viral sexually transmitted infection, does not present nowadays an efficient and safety therapy. To improve the healing, authors propose the use of diphenyl diselenide ((PhSe)2), a compound with already confirmed action against protozoa. For exerting its biological potential for a vaginal adminsitration (PhSe)2 needs to be incorporate in a nanocapsules suspension in mucoadhesive hydrogels. Hence, the focus of the work is the characterization of this drug delivery system based on a locust bean gum hydrogel loaded with a (PhSe)2 nanocapsules suspension.

Authors compare the biological action of free and nanoencapsulated (PhSe)2, and investigate the effect of drug loaded on hydrogel rheological behavior. The complete system is then evaluated for the mucoadhesive properties, the permeation efficiency and the irritant potential.

Encouraged results indicate this formulation as very promising for the treatment of Trichomonas vaginalis infection.

Data presentation is clear and well described.

Rheological characterization is very important to the practical final application. In cervix there is the presence of moisture, lubricative mucus and surely shear forces. Was HG viscosity evaluated at the real shear rate present in the vaginal area?

Experimental data verified the desired increment in mucoadhesive properties of HG-(PhSe)2 loaded, but authors didn’t justify which mechanisms lead to an higher bioadhesive strength. Have they any hypothesis?

Author Response

(The authors gave the same response as above.)

Round 2

Reviewer 1 Report

The author has extensively revised the manuscript and has improved the quality for publication. I recommend for publication.